# The role of playgrounds in the development of children's fundamental movement skills: A scoping review

Charlotte Skau Pawlowski[1,2]*, Cathrine Damsbo Madsen[1,2], Mette Toftager[1,2], Thea Toft Amholt[3], Jasper Schipperijn[1,2]

1 Department of Sports Science and Clinical Biomechanics, Research Unit for Active Living, University of Southern Denmark, Odense, Denmark, 2 World Playground Research Institute, University of Southern Denmark, Odense Denmark, 3 Center for Clinical Research and Prevention, Frederiksberg Hospital, Frederiksberg, Denmark

* cspawlowski@health.sdu.dk

**Data Availability Statement:** All relevant data are within the paper and its Supporting information files.

## Abstract

Fundamental movement skills (FMS) are the basic skills children should develop but are low in children from high-income countries. Literature indicates that playgrounds can play an important role challenging children's balance, agility, and coordination. However, knowledge on the influence of playgrounds on children's FMS development is fragmented. The aim of the present scoping review was to create an overview of all research that is relevant when studying the influence of unstructured playground play on children's FMS. Four electronic databases (Scopus, Web of Science, SportDiscus, and PsycInfo) were searched systematically in May 2022 and October 2023 following the PRISMA guidelines, leading to a final set of 14 publications meeting the inclusion criteria. The results of these publications indicate that it is important to design playgrounds with various features targeting balance, climbing, throwing, and catching to provide opportunities for children to enhance each FMS (i.e., stability, locomotor skills, and object control skills). Also, spreading features over a large area of the playground seems to ensure ample space per child, stimulate children to use locomotor skills by moving to and from features, and to play active games without equipment. Possibly, also natural play settings develop children's FMS. These findings, however, should be read with caution. More experimental studies using objective and standardized FMS tests are needed in this research field for a more robust conclusion.

## Introduction

Fundamental movement skills (FMS) are the basic skills children should be competent at, such as stability skills (e.g., sitting, standing, balancing), locomotor skills (e.g., running, jumping, climbing), and object control skills (e.g., throwing, catching) [1]. Preschool years are crucial in terms of developing various FMS [2]. Developing proficiency in these skills is important, as FMS provide an underlying base for successful participation in physical activities across the life course [1,3,4]. Despite the importance of FMS for physical activity, FMS have previously been shown to be low in preschool and school aged children from high-income countries [5].

**Funding:** Our study is funded by Kompan A/S who has funded the World Playground Research Institute at University of Southern Denmark. The authors Charlotte Skau Pawlowski, Mette Toftager, and Jasper Schipperijn are employed part time in the World Playground Research Institute and receive 50% of their salary from Kompan A/S and 50% of their salary from University of Southern Denmark. Cathrine Damsbo Madsen is employed fulltime in the World Playground Research Institute and receive 100% of her salary from Kompan A/S. The funders had no role in study design, data collection and analysis, decision to publish, or preparation of the manuscript.

**Competing interests:** We declare that no competing interests exist.

A recent systematic review showed that children (3–10 years old) in 25 countries were not achieving the FMS competence required to successfully participate in physical activity [6]. The ability to perform and master different types of FMS has also been associated with school readiness and performance [7,8], social interaction with peers [9], and self-perception [10].

To increase FMS, many interventions have been developed in various contexts [11–13], but few FMS interventions have been implemented at scale [14]. Adult-directed, structured FMS programs are considered effective in developing children's FMS [15]. However, structured programs require educated staff and thus are expensive to conduct. Further, specific training only affects the development of the specific task trained and not necessarily other tasks related to the same FMS competence [16].

Literature indicates that unstructured play at playgrounds is important for children's physical development, challenging their movement abilities such as balance, agility, coordination, and spatial awareness [17,18]. The World Health Organization claims that active play and opportunities for unstructured physical activity can contribute to the development of motor skills in children under five years [19]. However, playgrounds can be designed very differently, vary in size, features, and be built in conventional or natural materials. Further, studies addressing the influence of playgrounds on children's FMS development are carried out in different research fields with varying study designs, methods, and outcomes. A systematic review of active play interventions aimed at promoting physical activity and FMS conducted in 2016 only included four studies [20], and could not draw firm conclusion due to the small number of eligible studies and their heterogeneity. Thus, knowledge on the role of playgrounds in children's FMS development is fragmented and a coherent overview of relevant research in this field is needed to help guide future FMS interventions [21,22].

The aim of the present scoping review was to create a global, interdisciplinary overview of relevant research investigating the influence of unstructured playground play on children's FMS. More specifically we investigated the impact of playground size, playground setting, and playground features, respectively, on children's FMS. We did this investigation to be able to identify and outline existing knowledge in this research field and hereby identify knowledge gaps and set the agenda for future research.

## Materials and methods

The scoping review was conducted in accordance with the JBI methodology for scoping reviews [23] and the PRISMA guidelines were followed [24]. The review protocol was registered at Open Science Framework (https://doi.org/10.17605/OSF.IO/UYN2V) in May 2022.

The current scoping review is part of a broader project synthesizing evidence on the relationship between unstructured playground play and physical, mental, and social health among children and adolescents. Findings for other health outcomes than FMS when using playgrounds will be presented in separate articles.

### Search strategy

Four electronic databases, Scopus, Web of Science, SportDiscus, and PsycInfo, were searched systematically in May 2022. Search terms were tested and revised by all authors (having expertise in the research field) in collaboration with a research librarian from the University of Southern Denmark. It was decided to create a comprehensive search strategy with two search blocks to obtain all relevant research on the topic. One search block contained all identified synonyms for 'playground', the other contained all identified synonyms for 'children'.

To be fully updated on the literature when writing this article, we did a second search in the same four databases in October 2023. We search literature published in 2022–2023. This time

**Table 1. Search terms for Scopus.**

(TITLE (playground*)) OR (((TITLE-ABS (schoolyard* OR "school ground*") OR AUTHKEY (schoolyard* OR "school ground*")) OR (TITLE-ABS (play W/3 (area* OR space* OR environment* OR field* OR natural OR nature OR outdoor OR place* OR structure* OR equipment OR park*)) OR AUTHKEY (play W/3 (area* OR space* OR environment* OR field* OR natural OR nature OR outdoor OR place* OR structure* OR equipment OR park*))) OR (TITLE-ABS ((school* OR daycare* OR "day care" OR childcare OR "child care" OR kindergarten*) W/6 (play OR playable OR played OR playing OR "physical* activ*" OR "organi?ed activit*" OR "unorgani?ed activit*" OR "structured activit*" OR "unstructured activit*" OR "recreation* activit*" OR "leisure activit*" OR "outdoor activit*" OR "vigorous activit*")) OR AUTHKEY ((school* OR daycare* OR "day care" OR childcare OR "child care" OR kindergarten*) W/6 (play OR playable OR played OR playing OR "physical* activ*" OR "organi?ed activit*" OR "unorgani?ed activit*" OR "structured activit*" OR "unstructured activit*" OR "recreation* activit*" OR "leisure activit*" OR "outdoor activit*" OR "vigorous activit*"))) OR (TITLE-ABS (playfield* OR playplace* OR playscape* OR playspace* OR "public open space") OR AUTHKEY (playfield* OR playplace* OR playscape* OR playspace* OR "public open space")) OR (INDEXTERMS (playground)) OR (ABS (playground*) OR AUTHKEY (playground*))) AND ((TITLE-ABS (adolescen* OR baby OR boy OR schoolboy* OR boyhood OR girlhood OR child* OR schoolchild* OR girl OR schoolgirl* OR infan* OR juvenil* OR kid OR minor OR newborn* OR new-born* OR paediatric* OR pediatric* OR preschool* OR puber* OR pubescen* OR teen* OR tween* OR toddler* OR youth* OR student* OR schoolage*) OR AUTHKEY (adolescen* OR baby OR boy OR schoolboy* OR boyhood OR girlhood OR child* OR schoolchild* OR girl OR schoolgirl* OR infan* OR juvenil* OR kid OR minor OR newborn* OR new-born* OR paediatric* OR pediatric* OR preschool* OR puber* OR pubescen* OR teen* OR tween* OR toddler* OR youth* OR student* OR schoolage*)) OR (INDEXTERMS (child) OR INDEXTERMS (adolescent) OR INDEXTERMS (pediatric)))) AND ((TITLE-ABS ("Motor skill*")) OR (TITLE-ABS ("Motor competence")) OR (TITLE-ABS ("Motor planning")) OR (TITLE-ABS ("Motor fitness")) OR (TITLE-ABS ("Motor development")) OR (TITLE-ABS ("Locomotor skill*")) OR (TITLE-ABS ("Object control skill*")) OR (TITLE-ABS ("Gross-motor")) OR (TITLE-ABS ("Perceptual-motor competence")) OR (TITLE-ABS ("Perceptual-motor behavior")) OR (TITLE-ABS ("Perceptual-motor behaviour"))) OR (AUTHKEY ("Motor skill*" OR "Motor competence" OR "Motor planning" OR "Motor fitness" OR "Motor development" OR "Locomotor skill*" OR "Object control skill*" OR "Gross-motor" OR "Perceptual-motor competence" OR "Perceptual-motor behavior" OR "Perceptual-motor behaviour")))

we added one more search block to the two existing search blocks to specify our search around FMS as the health outcome investigated. The third search block contained all identified synonyms for 'fundamental movement skills'. In Table 1, the search terms for all three search blocks are shown for Scopus. The search terms were slightly adapted to fit each database.

## Selection criteria

For studies to be included in the current scoping review, they needed to take place on a playground. Playgrounds were defined as places designed or designated to facilitate play. We included publicly available outdoor playgrounds e.g., in parks or neighborhoods, as well as playgrounds at early childhood education and care (ECEC) centers, schools, and healthcare centers. We included all studies examining children aged 0–17 years regardless of the children's health condition and physical abilities. Studies were also included if the participants were parents or professionals around children. We included studies using all forms of study designs. Studies that were peer-reviewed, published from January 2000 to October 2023, and written in English were included. There was no limit on country or origin of studies. Further, in the present scoping review, studies were only included if FMS was one of the outcomes measured.

Studies were excluded if they only examined sports facilities (e.g., soccer fields, parkour parks, basketball courts, beach volley) and portable playground features (e.g., balls or socio-dramatic props such as teacups). Further, studies only focusing on adult-led playground activities such as physical education, organized activities, or supervision on the playground were excluded as well as studies about playground policy.

## Selection procedure

All references were imported to Endnote 20.0.1 where duplicates were removed by one researcher (CM) and uploaded to Covidence. Then all authors screened the publications' title and abstract (Jun-Aug 2022) whereafter each full-text was assessed by two of the authors independently (Aug-Oct 2022). Conflicts were solved by one of two authors (CSP or JS). The reference list of 10 randomly chosen included publications were screened by one author (CM) for additional publications of relevance (Jan 2023). Due to the comprehensive search strategy, this did not result in additional publications being included. Therefore, we did not check all reference lists for included publications.

For the present review, all full-text publications investigating the association between unstructured playground play and FMS were selected. Relevant citations in these publications were screened but no additional publications of relevance were found. Then after the second search, one author (CSP) screened the new publications' title and abstract whereafter full-text was assessed by two of the authors (CSP and CM) independently (Oct 2023). The results of the search and the study inclusion process are presented in a flowchart, Fig 1.

## Data extraction

For the current review, data extraction of the selected publications with FMS as a health outcome was completed by the first author (CSP) and cross-checked by MT. Data extracted included the aim of the study, study design, participants, setting, methods used to measure

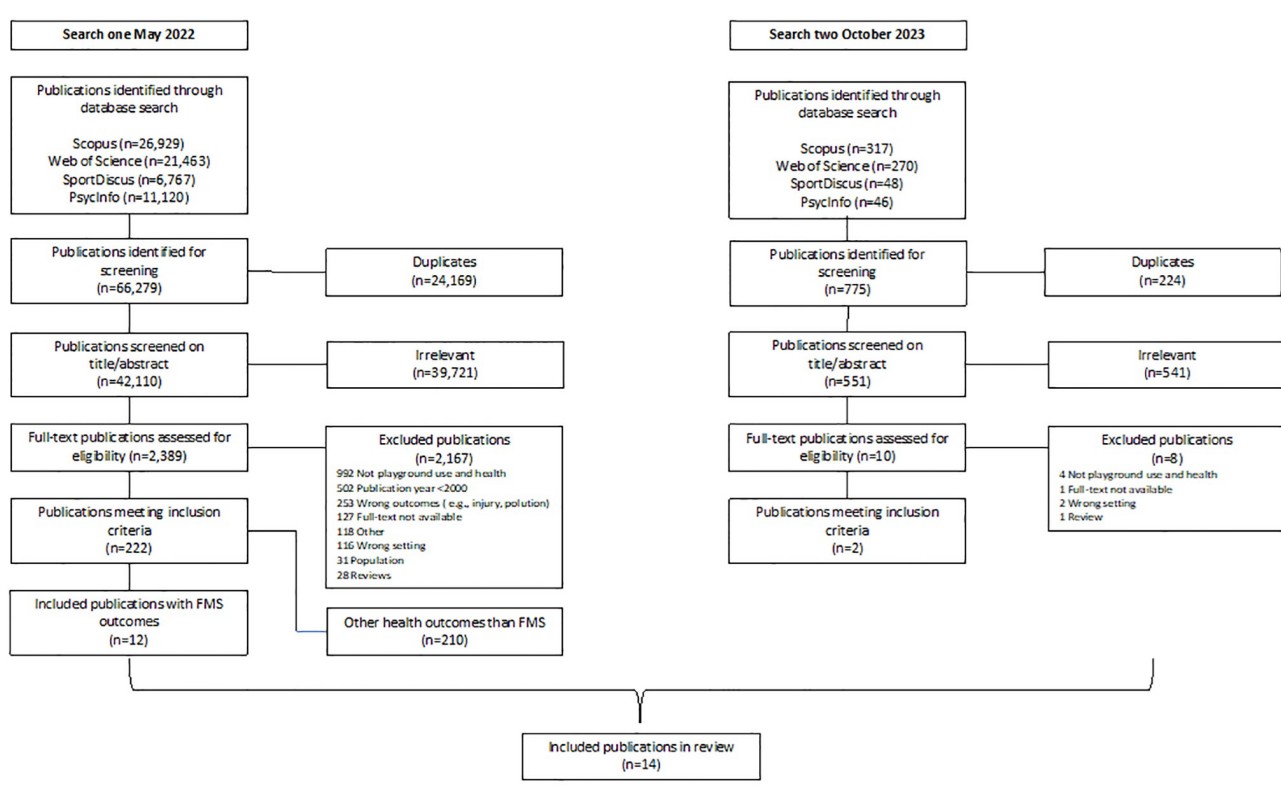

**Fig 1. Flowchart for selected publications.**

FMS, and key findings relevant to the review, as well as general information about the study such as author, country, and year of publication.

## Results

As seen in the flowchart (Fig 1), the total number of hits for search one was 66,279 related to the broad search on physical, social, and mental health outcomes in relation to children's playground use. After duplicates were removed 42,110 publications remained, of which 39,721 irrelevant titles and abstracts were excluded leaving 2,389 full-text publications to be screened. 1,941 full-text publications were excluded with reasons detailed in Fig 1. A total of 222 publications met the inclusion criteria, of which, 12 included FMS as at least one of the health outcomes investigated. For search two the total number of hits was 775 of which we removed 224 duplicates. From the 551 remaining publications, 541 irrelevant titles and abstracts were excluded leaving 10 full-text publications to be screened. Eight full-text publications were excluded with reasons detailed in Fig 1. A total of 2 publications from search two met the inclusion criteria. Thus, in total 14 publications were included in the current scoping review. Extracted data from the 14 included publications can be found in Table 2.

### Characteristics of the studies

Three studies were from the USA [25–27]. Two studies each were from Australia [28,29], Norway [30,31], and Italy [32,33]. The remaining studies were from England [21], Canada [34], Indonesia [35], Brazil [36], and Spain [37] (one study per country).

Half of the studies were conducted in early childhood education and care (ECEC) centers (n = 7) [21,25,26,30,31,35,36]. Four studies were conducted in a public area (i.e., neighborhood, park, sports center) [29,32,33,37]. Two studies were conducted at rehabilitation centers for children [27,34], and one study in primary schools [28].

Twelve studies included children as participants, except one study including parents [37] and one study including parents and ECEC staff members [31]. These two studies focused on adults' perceptions on children's FMS. Most of the child studies included a population of traditionally developed children (n = 10) [21,25,26,28–30,32,33,35,36]. One study included children with Down syndrome [34] and one study included both typically developed children and children with special needs that have led to atypical development [27]. The age of the children studied varied widely across the studies included. Six of the studies included children from the age of three; 3–5 years (n = 4) [21,25,26,36], 3–6 years (n = 1) [35], and 3–15 years (n = 1) [27]. One study included children from 4–6 years [33]. Four studies included children from the age of five; 5 years (n = 1) [32], 5–7 years (n = 1) [30], 5–10 years (n = 1) [29], and 5–12 years (n = 1) [28]. One study investigated children aged 6–7 [34]. Socioeconomic status (SES) was infrequently reported and none of the studies had a focus on low SES population.

Eleven study designs were cross-sectional without control groups [21,25–29,31,34–37]. Three studies were intervention studies, all with experimental group and control group [30,32,33]. All studies used quantitative methods. However, one study was a mixed methods study using qualitative adult interviews in combination with surveys [31].

Nine of the studies used test protocols to measure FMS, of which three studies used the Champs Motor Skills Protocol (CMSP) [21,25,26], one study used Pictorial Scale of Perceived Movement Skill Competence (PMCS) [35], one study used the Motor Fitness Test of the European Test of Physical Fitness (EUROFIT) [30], one study used the Test of Gross Motor Development (TGMD-2) [36]. Also, one study used the Movement Assessment Battery for Children (MABC-2), the Test of Motor Competence (TCM), and two playground balance tests [33]. Two studies used selected items of validated FMS tests such as sprint run, vertical jump, side

**Table 2. Data extraction of the included publications.**

| Author, publication year, and country | Study aim and design | Study setting and population | Measurement and results |
|---|---|---|---|
| Adams et al (2018), Australia [29] | **Aim:** Whether playground design facilitated different levels of PA and FMS<br>**Design:** Cross-sectional | **Setting:** Neighborhood (3 playgrounds in 1 city; traditional, adventure, contemporary).<br>**Population:** 57 children, 5–10 yrs. | **Measurement:** SOFIT (modified)<br>**Results:** There were no significant associations with FMS between the three playgrounds.<br>The most frequent performed FMS were locomotor skills (31.3%), specifically walking (18.3%) and running (11.3%). Body management skills (15.2%) and climbing (12.3%) was also observed at all three playgrounds, whereas object control skills such as catching and throwing were rarely observed (0.0–0.2%)<br>Children performed few FMS but used a wider variety of equipment in the contemporary and adventure playgrounds. |
| Famelia et al (2018), Indonesia [35] | **Aim:** Relationships among FMS, playground PA, and gender<br>**Design:** Cross-sectional | **Setting:** 4 ECEC centers (2 urban—2 rural)<br>**Population:** 66 children, 3–6 yrs. | **Measurement:** PMSC + the Test of Gross Motor Development-3<br>**Results:** No main effect of location for locomotor skills and perceived movement skill competence. |
| Fjørtoft (2001), Norway [30] | **Aim:** How playing in the natural environment might stimulate FMS<br>**Design:** Intervention (play in forest versus traditional playground 1–2 hours/day for 9 months) | **Setting:** 3 ECEC centers (1 experimental—2 control)<br>**Population:** 75 children, 5–7 yrs. (46 in the experimental group) | **Measurement:** EUROFIT + Beam walking and Indian skip<br>**Results:** At the posttest 9 months after the pretest, significant differences were found in eight out of nine FMS test items in the experimental group (flamingo balance ($p<0.001$), plate tapping ($p<0.001$), standing broad jump ($p<0.001$), bent arm hang ($p<0.001$), Indian skip ($p<0.001$), sit-ups ($p<0.01$), beam walking ($p<0.01$), and shuttle run ($p<0.01$)) whereas the control group experienced a significant difference in three test items (standing broad jump ($p<0.01$), bent arm hang ($p<0.001$), and Indian skip ($p<0.001$).<br>At the pretest the experimental group scored lower than the control group, but scored better in all test items at the posttest |
| Foweather et al (2021), England [21] | **Aim:** The association between play behavior and FMS during recess at preschool<br>**Design:** Cross-sectional | **Setting:** 12 ECEC centers<br>**Population:** 133 children, 3–5 yrs. | **Measurement:** Video-assessment using CMSP<br>**Results:** Relative to time spent in other types of play behaviors, time spent in play without equipment was positively associated with total FMS and locomotor skills, while time spent in locomotion activities (moving while not engaged in an active play game) was negatively associated with total FMS and locomotor skills |
| Gil-Madrona et al (2019), Spain [37] | **Aim:** The contribution of public playgrounds to obesity reduction, motor, social, and creative development<br>**Design:** Cross-sectional | **Setting:** Neighborhood (41 parks in 1 city)<br>**Population:** 1019 adults | **Measurement:** Survey<br>**Results:** 53.7% parents agreed with the positive contribution of public playgrounds to motor skills (38% quite agree and 15.7% totally agree). |
| Grunseit et al (2020), Australia [28] | **Aim:** Relationship between school playground size and PA, fitness, and FMS<br>**Design:** Cross-sectional | **Setting:** 43 primary schools<br>**Population:** 5238 children, 5–12 yrs. | **Measurement:** Scoring of 7 FMS skills<br>**Results:** No association between playground space and motor skills. |
| Loftesnes (2021), Norway [31] | **Aim:** Evaluating a new-built nature playground for children aged 2–6 years<br>**Design:** Cross-sectional post-intervention study (A nature playground used in one year for min. 2 days a week) | **Setting:** 9 ECEC centers<br>**Population:** 30 adults (18 staff + 12 parents) | **Measurement:** Survey and interview<br>**Results:** Parents found their child being more able to cope with motor skills. |

*(Continued)*

**Table 2.** (Continued)

| Author, publication year, and country | Study aim and design | Study setting and population | Measurement and results |
|---|---|---|---|
| Miller et al (2017), USA [27] | **Aim:** Quantify equipment/areas impacted for children with sensory challenges<br>**Design:** Cross-sectional | **Setting:** Child rehabilitation center (1 playground)<br>**Population:** 181 children, 3–15 yrs. (41 with special needs) | **Measurement:** Coding of camera recordings<br>**Results:** The behavior most often observed across all pieces of equipment was novel use, ranging from 41.82–97.66% of the time. Least = Mobius, most = sand and water. Motor planning was highest for the Mobius Climber (58.18%) and lowest for sand and water. |
| Nobre et al (2023), Brazil [36] | **Aim:** factors associated with FMS in preschoolers from a Brazilian urban area<br>**Design:** Cross-sectional | **Setting:** 9 ECEC centers<br>**Population:** 211 children, 3–5 yrs. | **Measurement:** TGMD-2<br>**Results: Children** from ECEC centers with a park or courtyard had significant higher locomotion skills than those who did not have outdoor spaces in the ECEC center. |
| Szeszulski et al (2022), USA [25] | **Aim:** Association between the characteristics of the ECEC center environment and FMS<br>**Design:** Cross-sectional | **Setting:** 16 ECEC centers<br>**Population:** 172 children, 3–5 yrs. | **Measurement:** CMSP<br>**Results:** Better outdoor play environment quality score and more outdoor equipment were positively associated with higher CMSP scores. |
| Tortella et al (2016), Italy [32] | **Aim:** Effects of structured and unstructured activities played at the playground on FMS<br>**Design:** Intervention (10 weeks– 1 hour/week—half structured and half unstructured play at specific playground) | **Setting:** Neighborhood (1 playground)<br>**Population:** 110 children, 5 yrs. (71 in the experimental group) | **Measurement:** Scoring of 9 FMS skills (3 for fine and 6 for gross motor skills)<br>**Results:** The experimental group improved significantly in 4 out of 6 gross motor tasks (putting a medicine ball (p<0.001), one leg balance on left foot (p<0.05), balance on beam (p<0.001), and balance on platform (p<0.001)) and in none of the fine motor tasks. |
| Tortella et al (2022), Italy [33] | **Aim:** Effects of partly structured activities or free play on FMS<br>**Design:** Intervention (10 weeks– 1 hour/week–partly structured versus free play at specific playground) | **Setting:** Neighborhood (1 playground)<br>**Population:** 141 children, 4–6 yrs. (62 in experimental group 1; partly structured, 43 in experimental group 2; free play, 36 in control group) | **Measurement:** MABC-2 + TCM + Balance on Elastic Platforms Task and Balance on Beam Task<br>**Results:** No significant difference in motor competence measured by the TCM or the MABC-2 between groups. A significant improvement was found in the Platforms Task and Balance on Beam Task in the partly structured activity group compared to the free play and control groups. |
| True et al (2017), USA [26] | **Aim:** The contribution of various preschool environmental characteristics to children's FMS<br>**Design:** Cross-sectional | **Setting:** 22 ECEC centers (4 head start, 7 faith-based, 11 commercial)<br>**Population:** 229 children, 3–5 yrs. | **Measurement:** CMSP<br>**Results:** Playground size is a significant predictor of total motor score (effect size 0.33) when adjusting the analyses for other significant predictors, e.g., age, classroom size, teacher education and electronic media use but not locomotor score and object control score, individually.<br>Time spent in outdoor open spaces, fixed and portable playground equipment were non-significant predictors to total gross motor scores. |
| Virji-Babul et al (2006), Canada [34] | **Aim:** Analyzing the level of motor engagement within the playground<br>**Design:** Cross-sectional | **Setting:** 1 Child rehabilitation center<br>**Population:** 6 children with DS, 6–7 yrs. | **Measurement:** Coding of camera recordings<br>**Results:** Children spent a great amount of time in motor-based activities (90%) in a playground setting. The primary motor activity was swinging. The tasks appeared to become more difficult as the environment became more complex (even surface versus grass and incline surface). |

CMSP = the Champs Motor Skill Protocol; DS = Downs syndrome; ECEC = Early childhood education and care; EUROFIT = European Test of Physical Fitness, the Motor Fitness Test; FMS = fundamental movement skills; MABC-2 = Movement Assessment Battery for Children; PA = physical activity; PMSC = Pictorial Scale of Perceived Movement Skill Competence; yrs. = years; SOFIT = System for Observing Fitness Instruction Time; TGMD-2 = Test of Gross Motor Development; TMC = Test of Motor Competence.

gallop, one leg balance, heel-to-toe walking, catch, and putting a medicine ball [28,32]. Two studies each coded camera recordings [27,34], and used survey [31,37]. Loftesnes [31] combined survey and interview. One study used the systematic observation method SOFIT [29].

## Size of playground and fundamental movement skills

Two studies investigated the association between FMS and playground size in ECEC centers and primary schools, respectively. The cross-sectional study by True et al. 2017 including 229 children aged 3–5 from 22 ECEC centers in USA found a small positive relationship (effect size 0.33) between children's overall FMS competence and ECEC center playground size measured by Champs Motor Skills Protocol. Playground size was significantly associated with total motor score but not locomotor score and object control score individually [26]. In contrast, Grunseit et al. (2020) found no association between playground size and FMS when adjusting for relevant covariates in a cross-sectional study testing 210 children aged 5–12 in 43 Australian primary schools.

## Playground setting and fundamental movement skills

Four studies investigated FMS in relation to the setting of the playground. In a cross-sectional study conducted in two urban and two rural ECEC center playgrounds in Indonesia including 66 3-6-year-old children, Famelia et al. [35] found no main effect of rural versus urban ECEC center playgrounds for locomotor skills and perceived movement skill competence even though children at the rural ECEC center playgrounds were found to be more sedentary than children in the urban ECEC center playgrounds. Another cross-sectional study was conducted in 9 ECEC centers in an urban area in Brazil including 211 3-5-year-old children. In this study it was found that the children from ECEC centers with a park or courtyard had significant higher locomotion skills than those who did not have outdoor spaces in the ECEC center [36]. Oppositely, the two Norwegian studies investigated the association between play in rural areas and FMS in a ECEC setting. The one study was a post-intervention study evaluating newly built nature playgrounds in nine ECEC centers. After being used for minimum two days a week in a one-year period, 12 parents being interviewed experienced their 2-6-year-old children being more able to cope with motor skills [31]. Fjørtoft [30] conducted an intervention study investigating 5-7-year-old children's FMS after playing in a forest for 1–2 hours per day (experimental group of 46 children from one ECEC center) versus playing in a traditional ECEC center playground for 1–2 hours per day (control group of 29 children from two different ECEC centers). At the post-test 9 months after the pretest, significant differences were found in eight out of nine FMS test items in the experimental group (flamingo balance ($p<0.001$), plate tapping ($p<0.001$), standing broad jump ($p<0.001$), bent arm hang ($p<0.001$), Indian skip ($p<0.001$), sit-ups ($p<0.01$), beam walking ($p<0.01$), and shuttle run ($p<0.01$)) whereas the control group experienced a significant difference in three test items (standing broad jump ($p<0.01$), bent arm hang ($p<0.001$), and Indian skip ($p<0.001$)). Thus, the motor fitness test showed a general tendency that the children using the forest as a playscape performed better in a variety of motor skills than the children on the traditional playground [30].

## Playground features and fundamental movement skills

Eight studies investigated FMS in relation to playground features. Six of the studies included traditionally developed children in ECEC centers (n = 3) and public playgrounds (n = 3), whereas two studies included children on a playground in a child rehabilitation center. These two studies focused on children with special needs that have led to atypical development. The

one ECEC center study was a cross-sectional study from the USA including 172 3-5-year-old children conducted in 16 ECEC centers. They found that a higher-quality outdoor play environment (e.g., shade, number of play areas, bike path quality), and more outdoor play equipment were associated with higher locomotor skills measured using Champs Motor Skills Protocol [25]. In contrast, another cross-sectional study also using Champs Motor Skills Protocol to measure 133 3-5-year-old children from 12 UK ECEC centers found that time spent in active games without use of play equipment (e.g., chasing games/rough and tumble) was positively associated with higher total FMS and locomotor skills scores [21]. Active games with fixed equipment (e.g., climbing frame) or portable equipment (e.g., balls or socio-dramatic props such as teacups) were not associated with FMS in this study. Likewise, time spent in locomotion activities (i.e., moving while not engaged in an active play game) was negatively associated with total FMS and locomotor skills. In line with this, the third ECEC center study conducted among 229 3-5-year-old children in 22 ECEC centers also found that fixed and portable playground equipment were non-significant predictors to total gross motor scores when using Champs Motor Skills Protocol as an FMS measurement tool [26].

In a cross-sectional study from Spain, Gil-Madrona et al. [37] investigated the contribution of public playgrounds with classic features (such as slides, climbing frames, and swings) on children's FMS seen from a parent perspective. They found that 53.7% of the 1,019 parents included in the study agreed with the positive contribution of public playgrounds to motor skills. An Italian intervention study investigated a public playground designed with specific features to promote mobility, balance and manuality. They showed a significant improvement in the experimental group of 5-year-old children (n = 71) versus a control group of children (n = 39) in four out of six gross motor tasks (putting a medicine ball ($p<0.001$), one leg balance on left foot ($p<0.05$), balance on beam ($p<0.001$), and balance on platform ($p<0.001$)) after 30 minutes of structured play and 30 minutes of unstructured play once a week for a 10 weeks period [32]. In an Australian cross-sectional study, Adams et al. [29] investigated three different public playgrounds; a traditional, an adventure, and a contemporary public playground, in relation to FMS by systematically observing play in 57 children aged 5–10 years at the respective playgrounds. They found that children used a wider variety of features in the contemporary and adventure playgrounds compared to the traditional playground. However, no significant association with FMS between the three types of public playgrounds was found, possibly because a low amount of time in motor-based activities was observed. Still, the most frequently performed FMS were locomotor skills (31.3%), whereas object control skills were rarely observed (0.0–0.2%) at the three different public playgrounds [29].

Oppositely, coding of camera recordings in a Canadian cross-sectional study conducted at a rehabilitation center playground showed that six 6-7-year-old children with Down syndrome spent a great amount of time in motor-based activities (90%) in the playground setting. The primary motor activity was swinging. The tasks appeared to become more difficult as the environment became more complex (i.e., from even surface to grass and incline surface) [34]. In a cross-sectional study conducted at a rehabilitation center playground in USA among 181 3-15-year-old children both typically developed and with special needs atypically developed (n = 41), Miller et al. [27] coded camera recordings and found that novel use (i.e., ideation; child uses the equipment in a novel way), and motor planning (i.e., skilled, nonhabitual movements used to accomplish multistep tasks) were observed at all six playground features (sand and water table, jungle gym, Roller Slide, Mobius Climber, Cozy Dome, Omnispin Spinner). Novel use was observed most at a 'sand and water' table and least at 'Mobius Climber' (a climbing wall). In contrast, motor planning was highest for the Mobius Climber and lowest for the 'sand and water' table [27].

## Discussion

In the following, we will discuss our results on playground size, setting, and features and high-light how future playgrounds should be designed to support FMS development of children.

### Do we need large playgrounds?

Since studies have demonstrated that children are more active in large playgrounds [38,39], it seems obvious to conclude that more space provides more opportunities for FMS acquisition. In the study by Grunseit et al. [28], the authors found an association between the amount of playground space available and self-reported physical activity and objectively measured fitness, but interestingly they did not find an association between playground space and FMS. Given the strong predictive association between levels of physical activity and FMS competence [5] and the positive association between playground space and both physical activity and fitness showed in the study by Grunseit et al. [28], the reason for the lack of an association between playground space and FMS is unclear. However, in the study by True et al. [26], larger play-ground size was significantly associated with higher total FMS score. The reason could be that the age of the children in the two studies differed. In the study by Grunseit et al. [28], the chil-dren were 5–12 years old whereas in the study by True et al. [26], the children were 3–5 years old (preschool years) which is identified as a crucial time in terms of forming and developing FMS [5]. It is therefore possible that the children in the study by Grunseit et al. [28] had past the crucial time for developing FMS lowering the influence of playground size on FMS. In fact, in another study they found that 3–7 years-old children from rural areas with the lowest resi-dential density had better FMS than their peers from urban areas with the highest residential density [40]. Although the focus in this study was not specifically on playgrounds, Niemiströ et al. [40] concluded that because children spend multiple hours in ECEC centers, they believe that the size of the outdoor environment near these centers (such as playgrounds) plays a nota-ble role in children's motor development.

### Do we need nature playgrounds?

Jointly, the two Norwegian studies included in this review [30,31], indicated a positive impact of the natural environment on children's motor development. Further, the study by Nobre et al. [36] showed a positive effect of parks/courtyard in ECEC centers on children's locomotor skills. Similarly, a systematic review indicated some association between nature play and FMS even though this review did not focus specifically on playgrounds [41]. However, it is worth examining if the effect shown on green playgrounds is due to these playgrounds being placed in areas that might be larger and having a lower population density, as discussed above. In the study by Fjørtoft [30], the nature space used for playing by the experimental group of children was larger and herewith lower in population density than the traditional playgrounds in the ECEC centers used by the control groups. Also, in the study by Loftesnes [31], the natural space used for building a nature playground was larger and lower in population density than the traditional playground. In the study by Nobre et al. [36] size and population density was not mention but could be the explanation for effect since the ECEC centers with parks/court-yards were compared with ECEC centers without outdoor spaces at all. On the other hand, no effect of location was found in the study by Famelia et al. [35] investigating urban playgrounds in the city against rural playgrounds in farming areas in Indonesian ECEC centers. In this study, size or population density of the playgrounds were not mentioned, but it was described that limited space occurred at some settings, and they found children to be sedentary in the playgrounds around 70% of playground time, indicating that the playgrounds were relatively small. In line with this, a Norwegian study showed no differences in FMS competence of

children attending nature preschools and traditional preschools [42]. This could support that playground size and density have a greater impact on FMS than nature itself. However, we know too little about how the natural environment functions as a playground developing children's FMS to draw any conclusions on this topic.

## What features do we need on the playgrounds?

In the study by Szeszulski et al. [25], the authors found both number of features and quality of features in the ECEC centers' outdoor environment to influence children's locomotor skills. On the other hand, Foweather et al. [21] found that time spent in active games without equipment was positively associated with higher locomotor skills score and total FMS. This finding suggests that spending more time on active games such as dancing, chasing games, and rough and tumble play without use of playground features may be important for FMS development. Previous research has also demonstrated that preschool children in the highest locomotor skill tertile generally engaged in more dancing than children in the lowest tertile [43]. In the study by Foweather et al. [21], however, children spent a relatively large proportion of time (41%) engaged in active games with equipment, but this type of play was not associated with FMS, possibly because the children were frequently observed being sedentary on the equipment. It is possible that these pieces of equipment supported other FMS capacities, such as climbing or stability skills, not assessed in the study by Foweather et al. [21]. Nevertheless, this finding is similar to Adams et al. [29] reporting that the children used a wider variety of equipment in the contemporary and adventure playgrounds than the traditional playground, but they did not find a statistically significant association between the FMS observed at the three playgrounds varying in features. These authors suggest that it is possible that the general low FMS mastery among children could be influenced by the lack of FMS required to play in playgrounds [29].

From the studies, however, various features seem to encourage varying motor competences making it complex to answer exactly what features are needed in the playground to improve children's FMS development. In the study by Adams et al. [29], locomotor skills such as walking and running were observed most frequently in the contemporary playground where the features were spread over a large area requiring children to use locomotor skills to move around. Conversely, locomotor skills were observed less frequently at the adventure playground where the features were linked off a large walkway and children needed different FMS to move to and from different features such as balancing. Still, climbing nets were the most used play feature at the adventure playground stimulating locomotor skills [29]. Climbing and hanging features are also important to develop upper-body strength [44]. Importantly, Adams et al. [29] and True et al. [26] found no association between playground play and object control skills. According to True et al. [26], features to improve object control skills seem not to be provided very often in playgrounds for preschool children. In line with this, a study found object control skills to develop at a slow rate before the age of 9–10 [45]. Portable features such as balls might influence object control skills. Portable features, however, was not studied in the current review. As Tortella et al. [32] showed, specifically targeted playground equipment may be necessary to encourage FMS development. The authors conclude, however, that specific training using specific playground features, only affects the development of that task trained and not necessarily other tasks related to the same FMS competence [32]. This statement was supported in the other study by Tortella et al. [33]. In this study structured activity at the FMS specific playground was found to improve children's balance more than free play at the same playground. In line with this, Revie and Larkin [16] found a significant effect of eight sessions

of intensive teaching of FMS in children with poor coordination. Though, this study did not take place on a playground.

A sensory-rich playground provided with varied features, enticing colors, and multitextured materials seem to be valuable for the development of children with disabilities [27]. However, according to Virji-Babul et al. [34] children with special needs atypically developed seem to have more difficulties in extracting and processing relevant information from the physical environment than children typically developed, leading to decreased engagement in free play at the playground. Thus, it seems important that playground features are easy to interpret and can be used at different developmental stages.

## Strengths and limitations

We followed the JBI methodology and PRISMA guidelines for scoping reviews for a robust, rigorous, and transparent review protocol, thus the risk of bias in our review methodology is low [23]. Further, a strength is that the search procedure was developed by a research group of experts in the research field of playground usage in collaboration with a librarian with huge expertise in search strategies. To capture as much relevant research, four different databases were searched. However, given the large number of publications retrieved, we questioned if we should have created a third block containing health outcomes to narrow-down our search. That was also the reason why we added a third search block in our second search. Further, no quality assessment of included publications was performed. Since only 14 publications were included in the present scoping review, we wanted to cover all knowledge on the subject regardless of the design and quality of the study. A challenge was that the studies used many different child-monitoring instruments to measure FMS, possibly because there is little agreement on what FMS measurement should be used [46].

## Conclusion and future directions

The aim of the current scoping review was to create an overview of all research that is relevant when studying the influence of unstructured playground play on children's FMS. Fourteen studies investigated unstructured playground play and children's FMS. From the current scoping review, it seems important to design playgrounds with various features targeting balance, climbing, throwing, and catching to provide opportunities for children to enhance each FMS (i.e., stability, locomotor skills, and object control skills). Further, spreading features over a large area seems to both ensure ample space per child and to stimulate children to use locomotor skills by moving to and from features and by playing active games without equipment. Possibly, also natural play settings develop children's FMS. Our results, however, should be read with caution. Overall, based on only 14 studies reviewed, we still know too little about the association between unstructured playground play and FMS, and more effort should be dedicated to future studies in this field. In particular, we need more experimental studies using standardized FMS tests since only three of the 14 studies had this high-quality design. Also, it is needed to discuss the quality of the used FMS tests in future research.

## Supporting information

**S1 Checklist. Preferred Reporting Items for Systematic reviews and Meta-Analyses extension for Scoping Reviews (PRISMA-ScR) checklist.**
(DOCX)

## Acknowledgments

We thank academic officer Danielle Nørager Johansen from University of Southern Denmark who helped organizing and coordinating the search and extraction process. Also, thanks to librarian Lasse Østergaard from University of Southern Denmark who helped with the search strategi.

## Author Contributions

**Conceptualization:** Charlotte Skau Pawlowski.

**Data curation:** Charlotte Skau Pawlowski, Cathrine Damsbo Madsen, Mette Toftager.

**Project administration:** Charlotte Skau Pawlowski.

**Software:** Cathrine Damsbo Madsen.

**Validation:** Mette Toftager.

**Writing – original draft:** Charlotte Skau Pawlowski.

**Writing – review & editing:** Cathrine Damsbo Madsen, Mette Toftager, Thea Toft Amholt, Jasper Schipperijn.

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
