## [Decision Letter · Decision Letter 0]

16 Aug 2023

PONE-D-23-14269The role of playgrounds in the development of children’s fundamental movement skills: A scoping reviewPLOS ONE

Dear Dr. Pawlowski

Thank you for submitting your manuscript to PLOS ONE. After careful consideration, we feel that it has merit but does not fully meet PLOS ONE’s publication criteria as it currently stands. Therefore, we invite you to submit a revised version of the manuscript that addresses the points raised during the review process.

We look forward to receiving your revised manuscript.

Kind regards,

Renato S. Melo, PhD

Academic Editor

PLOS ONE

Journal Requirements:

3. Please address the following queries:

Additional Editor Comments :

Dear Dr. Charlotte Skau Pawlowski

Thank you for submitting your manuscript to PLOS ONE. After careful consideration, we feel that it has merit but does not fully meet PLOS ONE’s publication criteria as it currently stands. Therefore, we invite you to submit a revised version of the manuscript that addresses the points raised during the review process.

We look forward to receiving your revised manuscript.

Kind regards,

Renato S. Melo, PhD

Academic Editor

PLOS ONE

Reviewers' comments:

Reviewer's Responses to Questions

**Comments to the Author**

1. Is the manuscript technically sound, and do the data support the conclusions?

Reviewer #1: Yes

Reviewer #2: Yes

2. Has the statistical analysis been performed appropriately and rigorously? 

Reviewer #1: N/A

Reviewer #2: Yes

3. Have the authors made all data underlying the findings in their manuscript fully available?

Reviewer #1: Yes

Reviewer #2: Yes

4. Is the manuscript presented in an intelligible fashion and written in standard English?

Reviewer #1: Yes

Reviewer #2: Yes

5. Review Comments to the Author

Reviewer #1: This manuscript has been reviewed according to the PRISMA extension for Scoping Reviews (PRISMA-SCR). The main aim of this scoping review was to create an overview of research that is related to the influence of structured playground play on children’s FMS. The authors findings will be a valuable contribution to the paediatric research literature and assists to fill a gap in this literature and provide focus for future research by the authors and other researchers working in this space. I have provided line by line comments below and hope this feedback assists the authors to progress their manuscript through to publication.

Thank you for the opportunity to review this interesting and valuable manuscript.

Line 37 - This is a firm statement with only one reference to support the statement. Could this be rephrased to state something like: FMS have previously been shown to be low in preschool and school aged children from high-income countries.

Line 48 - For clarity, please replace the word 'this' with exactly what you are referring to.

Line 50 - The word 'also' is over used in this manuscript. Please consider editing the manuscript to reduce the number of times it is required.

Line 56 - Please insert 'systematic' in front of the word 'review'.

Line 58 - Please insert full-stop after the word 'heterogeneity'.

Line 61 - Please replace the word 'article' with 'scoping review'.

Line 62 - It appears in the way you have presented your results that you had some specific aims (e.g. impact of playground size, location etc). If this is correct, could you please more accurately state your aims?

Line 68 - Please confirm if you have utilised the PRISMA extension for scoping reviews (PRISMA-ScR).

Tricco, AC, Lillie, E, Zarin, W, O'Brien, KK, Colquhoun, H, Levac, D, Moher, D, Peters, MD, Horsley, T, Weeks, L, Hempel, S et al. PRISMA extension for scoping reviews (PRISMA-ScR): checklist and explanation. Ann Intern Med. 2018,169(7):467-473. doi:10.7326/M18-0850.

Line 76 - As this review is now over 12 months old, could you please update your search to ensure it is within a 6 month period of recency?

Line 109 - Currently you jump back and forth between inclusions and exclusions. Could you please reorder this content to have the inclusions and exclusions grouped together for improved clarity?

Line 154 - Please check Table numbers. At the end of the manuscript you list Table 1 (not 2) as the data extraction table.

Line 160 - Can you provide some rationale for your dissecting the publications by decades? Are the decades meaningful by decades (new rules / policy etc). Why is this information important for the reader to know about? How does it relate to your aims? You may need to put some information in your methods about this step of data synthesis.

Line 165 - Could you please rewrite these characteristics summaries to refer to studies. I.e., the publications were not undertaken in early childhood education centers but rather the studies were.

Line 168 - Here you state 2 studies included adults. Did they also include children 0-17? (As in your inclusion criteria you state you were including publications examining children aged 0-17 years). If this study did also included children 0-17 years, then in your methods, you will need to be clear that you will accept studies / publications where 0-17 year olds are included in addition to adults. However, if these two studies are just about adults, then according to your inclusion criteria you should exclude them. It is unclear to me if these two studies with adults in ref 35 and 31 or not?

Line 172 - Were the children actually disabled? Or did they have a condition / impairments that led to atypical development? Please consider the use of the word 'disabled' and if it is the correct use of this term (I am not sure, it may or may not be, but the use of this language should carefully considered in line with ICF language expectations).

Line 192 - I am not sure what is meant by this sentence. Loftesnes is not an English Language word (to my knowledge). Could you please help the English language readers to better understand the meaning of this sentence? ADDITIONAL STATEMENT: I think from reading the discussion this is actually an author (Please add the reference citation for the author).

Line 217 - Please correct to post-test

Line 228 - Please see statement above about the use of the word 'disabled' children. If the children were in fact 'disabled' please use person first language in line with expectations of ICF. E.g. Children with diagnosed conditions resulting in disability.

Not all children with impairments will be 'disabled'.

Line 235 - Are you able to provide more information about the 'play equipment' here...so that the reader can better interpret the findings?

Line 251 - Replace 'at' with 'a'

line 254 - insert years after 5-10.

Line 267 - As above for statement - change to person first language.

Lines 275 - 276: If this was not connected with your original aims, then I would encourage you to write something about these key themes prior to presenting them, in your results section.

Line 283 - Replace 'objective' with 'objectively'

Line 304 - Please replace 'worth to examine' with 'worth examining'

Line 364 - Please replace 'traditional' with 'traditionally'

Line 368 - Please provide some detail about the robust features of your methods.

Line 373 - This is a nice insight to share.

Line 531 - For Table 1 - Please consider reformatting so that it can be included in the manuscript. Currently the final right hand side column has most relevant content. You could merge the FMS headings and in the column put sub headings:

FMS Measurement:

FMS Result:

Also the same for

Study Aim:

Study Design:

and

Population:

Setting:

This way your table can more easily be included with the manuscript without a lot of empty space.

Reviewer #2: This is a scope review, which was very well designed by its authors and minor revisions were requested. After the corrections made by the authors in this second review, we observed that all the suggestions requested by the reviewers were accepted and carried out in the manuscript, which they did a good job. Therefore, the article is now ready to be accepted for publication.

6. PLOS authors have the option to publish the peer review history of their article (what does this mean?). If published, this will include your full peer review and any attached files.

Reviewer #1: No

Reviewer #2: No

---

## [Author Response · Author response to Decision Letter 0]

2 Sep 2023

Thank you for letting us correct and clarify the manuscript. We have responded to the editor comments in a rebuttal letter. We were a little unsure about what to do in relation to some of the comments - so let us know if we have misunderstand some requested corrections.

---

## [Decision Letter · Decision Letter 1]

24 Sep 2023

PONE-D-23-14269R1The role of playgrounds in the development of children’s fundamental movement skills: A scoping reviewPLOS ONE

Dear Dr. Pawlowski

Thank you for submitting your manuscript to PLOS ONE. After careful consideration, we feel that it has merit but does not fully meet PLOS ONE’s publication criteria as it currently stands. Therefore, we invite you to submit a revised version of the manuscript that addresses the points raised during the review process.

We look forward to receiving your revised manuscript.

Kind regards,

Renato S. Melo, PhD

Academic Editor

PLOS ONE

Journal Requirements:

Additional Editor Comments:

Dear authors, the modifications requested by one of the reviewers were not fully clarified by you, therefore, the reviewer requested that the suggestions be made and those that were not, that is, those that were refuted, that you justify in a response letter to the reviewers.

Reviewers' comments:

Reviewer's Responses to Questions

**Comments to the Author**

1. If the authors have adequately addressed your comments raised in a previous round of review and you feel that this manuscript is now acceptable for publication, you may indicate that here to bypass the “Comments to the Author” section, enter your conflict of interest statement in the “Confidential to Editor” section, and submit your "Accept" recommendation.

Reviewer #1: (No Response)

Reviewer #2: All comments have been addressed

2. Is the manuscript technically sound, and do the data support the conclusions?

Reviewer #1: Partly

Reviewer #2: Yes

3. Has the statistical analysis been performed appropriately and rigorously? 

Reviewer #1: N/A

Reviewer #2: Yes

4. Have the authors made all data underlying the findings in their manuscript fully available?

Reviewer #1: Yes

Reviewer #2: Yes

5. Is the manuscript presented in an intelligible fashion and written in standard English?

Reviewer #1: Yes

Reviewer #2: Yes

6. Review Comments to the Author

Reviewer #1: Dear Authors,

I appreciate the additional work you have completed to address the reviewer comments. I can see that you have addressed the comments I provided in the PRISMA checklist but I cannot see that you have addressed the comments (x 30 comments) in the original manuscript supplied by this reviewer on the first review. My apologies if this has somehow not reached you. I have attached the manuscript with comments again. Could I please ask that you address each comment line by line to demonstrate the response in a 'response to reviewer' letter and also highlight the changes made in the manuscript. I believe this is a valuable contribution to the field and look forward to seeing the author responses on your next submission.

Kind regards,

Reviewer.

Reviewer #2: I congratulate the authors for the changes, and I believe that this way the article is clearer for readers.

7. PLOS authors have the option to publish the peer review history of their article (what does this mean?). If published, this will include your full peer review and any attached files.

Reviewer #1: No

Reviewer #2: No

---

## [Author Response · Author response to Decision Letter 1]

7 Oct 2023

We have attached a rebuttal letter that responds to each point raised by the academic editor and reviewers

---

## [Decision Letter · Decision Letter 2]

30 Oct 2023

The role of playgrounds in the development of children’s fundamental movement skills: A scoping review

PONE-D-23-14269R2

Dear Dr. Pawlowski,

We’re pleased to inform you that your manuscript has been judged scientifically suitable for publication and will be formally accepted for publication once it meets all outstanding technical requirements.

Kind regards,

Renato S. Melo, PhD

Academic Editor

PLOS ONE

Additional Editor Comments (optional):

Reviewers' comments:

Reviewer's Responses to Questions

**Comments to the Author**

1. If the authors have adequately addressed your comments raised in a previous round of review and you feel that this manuscript is now acceptable for publication, you may indicate that here to bypass the “Comments to the Author” section, enter your conflict of interest statement in the “Confidential to Editor” section, and submit your "Accept" recommendation.

Reviewer #1: All comments have been addressed

Reviewer #2: (No Response)

2. Is the manuscript technically sound, and do the data support the conclusions?

Reviewer #1: Yes

Reviewer #2: Yes

3. Has the statistical analysis been performed appropriately and rigorously? 

Reviewer #1: N/A

Reviewer #2: Yes

4. Have the authors made all data underlying the findings in their manuscript fully available?

Reviewer #1: Yes

Reviewer #2: Yes

5. Is the manuscript presented in an intelligible fashion and written in standard English?

Reviewer #1: Yes

Reviewer #2: Yes

6. Review Comments to the Author

Reviewer #1: Thank you to each of the authors for making these additional changes. I feel they have contributed greatly to enhancing the paper for your audience. Congratulations on a great contribution of research to this important space. There is just one minor change that needs to be made (and could be made at proof checking), that is that your Scopus search terms is listed as table 2, when it is referenced as Table 1 in the text.

Reviewer #2: (No Response)

7. PLOS authors have the option to publish the peer review history of their article (what does this mean?). If published, this will include your full peer review and any attached files.

Reviewer #1: No

Reviewer #2: No

---

## [Editor Report · Acceptance letter]

20 Nov 2023

PONE-D-23-14269R2 

The role of playgrounds in the development of children’s fundamental movement skills: A scoping review 

Dear Dr. Pawlowski:

I'm pleased to inform you that your manuscript has been deemed suitable for publication in PLOS ONE. Congratulations! Your manuscript is now with our production department. 

Kind regards, 

on behalf of

Dr. Renato S. Melo 

Academic Editor

PLOS ONE